# Patient delay and associated factors among tuberculosis patients in Gamo zone public health facilities, Southern Ethiopia: An institution-based cross-sectional study

Asrat Arja[1]*, Wanzahun Godana[2], Hadiya Hassen[2], Biruk Bogale[3]

1 Bursa District Health Office, Bursa, Sidama Region, Ethiopia, 2 Schools of Public Health, Arba Minch University, Arba Minch, Ethiopia, 3 Departments of Public Health, Mizan Tepi University, Mizan Aman, Ethiopia

* asratarja1983@gmail.com

**Data Availability Statement:** All relevant data are within the paper and its Supporting Information files.

## Abstract

### Background

Delayed tuberculosis diagnosis and treatment increase morbidity, mortality, expenditure, and transmission in the community. Early diagnosis and initiation of treatment are essential for effective TB control. Therefore, the main objective of this study was to assess the magnitude and factors associated with patient delay among tuberculosis patients in Gamo Zone, Southern Ethiopia.

### Methods

A cross-sectional study was conducted in Gamo Zone, Southern Ethiopia from February to April 2019. Fifteen health facilities of the study area were selected randomly and 255 TB patients who were ≥18 years of age were included. Data were collected using a question-naire through face-to-face interviews and analyzed using SPSS version 20.0. Patient delay was analyzed using the median as the cut-off value. Multivariable logistic regression analysis was fitted to identify factors associated with patient delay. A p-value of ≤ 0.05 with 95% CI was considered to declare a statistically significant association.

### Results

The median (inter-quartile range) of the patient delay was 30 (15–60) days. About 56.9% of patients had prolonged patients' delay. Patient whose first contact were informal provider (adjusted odds ratio [AOR]: 2.24; 95% confidence interval [CI] 1.29, 3.86), presenting with weight loss (AOR: 2.53; 95%CI: 1.35, 4.74) and fatigue (AOR: 2.38; 95%CI: 1.36, 4.17) and body mass index (BMI) categories of underweight (AOR: 1.74; 95%CI: 1.01, 3.00) were independently associated with increased odds of patient delay. However, having good knowledge about TB (AOR: 0.44; 95% CI: 0.26, 0.76) significantly reduce patients' delay.

**Funding:** The author(s) received no specific funding for this work.

**Competing interests:** The authors have declared that no competing interests exist.

**Abbreviations:** TB, Tuberculosis; DOTS, Directly observed treatment short-course; HSTP, Health sectors transformation plan; PD, Patient delay; CBHI, Community-based health insurance; HCF, Health care facility; FIES, Food insecurity experience scale.

## Conclusion

In this study, a significant proportion of patients experienced more than the acceptable level for the patient delay. Knowledge about TB, the first action to illness, presenting symptoms, and BMI status were identified factors associated with patient delay. Hence, raising public awareness, regular training, and re-training of private and public healthcare providers, involving informal providers, and maintenance of a high index of suspicion for tuberculosis in the vulnerable population could reduce long delays in the management of TB.

## Introduction

Effective TB control globally depends on substantial changes in TB prevention and care strategies in countries with a high burden, including Ethiopia. However, the cornerstones of global TB control programs are the early identification and timely management of infectious TB cases successfully. Therefore, peoples who have had a cough for two weeks or more are encouraged to see health facilities for a diagnosis and early treatment [1, 2]. As a result, any delay in diagnosing and treating tuberculosis patients not only increases community spread but could also lead to a more advanced disease state, which can lead to more symptoms and a higher risk of death [3].

Tuberculosis has been recognized as a major global public health issue since 1993, and numerous global TB control measures have been developed and implemented, including Directly Observed Short-Course Treatment (DOTS), Stop TB, and End TB strategies [4–6]. Thus, successful diagnosis and treatment of TB saved an estimated 54 million lives between 2000 and 2017, and TB mortality has dropped by 33 percent since 1990. Given such progress and the fact that almost all cases can be treated, TB has remained one of the world's major public health concerns [7].

According to a recent WHO report, an estimated 10 million new cases of TB have occurred, of which only 6.4 million new cases of TB have been reported to national authorities and reported to the WHO, and most missing persons with TB are presumed to be receiving some form of treatment from public or private healthcare providers. This reflects a 3.6 million gap between the incident and notified cases. Ten countries accounted for 80 percent of this difference; India, Indonesia, and Nigeria were the top three, accounting for nearly half the gap (46%) [7].

In Ethiopia, tuberculosis has also been identified as a major public health problem and efforts to control it have begun since the early 1960s. However, TB has remained one of the country's major public health concerns, accounting for the third cause of hospital admission and the second cause of death [2]. With an estimated 219,186 new cases and 48,910 TB deaths, Ethiopia has been ranked among the 30 TB-HIV High Burden Countries (HBC) [8]. Moreover, according to a recent global report about 32% of TB cases from estimated new cases may not have been diagnosed and properly treated in Ethiopia [7]. These cases contribute to an increase in transmission, mortality, and morbidity and most transmissions occur from the onset of a cough to the initiation of treatment [9]. In addition, late diagnosing and treatment of TB patients has proven to be a major challenge to the TB control program, especially in countries with low incomes, including Ethiopia.

The government of Ethiopia has granted TB control due consideration, and prevention and control of TB and Leprosy are among the country's priority health programs in the country's

Health Sector Transformation Plan (HSTP) [10]. In Ethiopia, like most TB control programs, TB case finding were mainly relying on passive and community-based enhanced TB case finding as to the main strategy with focusing on diagnosing TB among people who actively seek medical care with TB symptoms or identified from the community through health extension workers. With this strategy, the program achieved to detect not greater than two-thirds of its annual estimated TB cases requiring an additional but efficient strategy to be implemented to achieve the ambitious targets set by END TB for 2035 [2, 10].

Factors contributing to patient delay (PD) include family size, occupation, the income of the family, stigma, knowledge about TB, first visit informal provider, distance to the health facility, and self-medication [11–14]. There are growing numbers of studies on diagnostic and treatment delays. However, little is known about the food insecurity status of the patient and its association with patient side delay [12, 15–17]. As a result, the identification and assessment of factors for patient delays has been set as one of the national priority research agenda [18]. There are growing pieces of evidence on the patient side of TB patients in Ethiopia. However, there have been no studies specifically in our research area, and recent information on treatment delay is highly needed. Furthermore, Diagnosis of TB and delay in treatment time and contributing factors to this delay differ across communities, types of health facilities visited, and geographical areas, including within population groups of the same local settings and disease category. This necessitates the conduct of localized studies to identify population-specific contributing factors to TB diagnosis and treatment delays. Therefore, this study aims to assess the magnitude and factors associated with patient delay among tuberculosis patients in Gamo Zone public health facilities.

## Materials and methods

### Study setting and design

An institution-based cross-sectional study was conducted from February 1 to April 1, 2019, in the Gamo Zone of the Southern Nations Nationalities and Peoples Region. The Zone is also located 500 km from Addis Ababa and 275km from Hawassa, the capital city of Southern Ethiopia. In this study area, there are 55 public health facilities (one general hospital, three primary hospitals, and fifty-one health centers), and forty-eight private clinics during the study period.

Diagnosis and treatment of all forms of TB across the country are based on the adopted national TB control guideline [2] that specific case definitions, diagnostic, and treatment standards. Public health institutions are the main sources of health care for most people: health services access, as defined by a residence within 10 kilometers of any health institution, is about 80% in the study area. Patients have free access to TB diagnostic services and treatment in public health facilities. Hence, all the districts are giving tuberculosis diagnosis and treatment services, including DOTS, at the patient's nearest potential health facilities, including health posts. The lowest level of a health facility is a health post, employed by two health extension workers (HEWs). In detecting and referring TB suspects to the next level of health care, which is the health center for diagnosis and initiation of treatment, HEWs play an important role.

### Sample size determination and sampling technique

The sample size was determined by using the formula required for the determination of sample size for estimating single population proportions considering the following assumptions: a proportion of patient delay (41.1%) taken from a previous study conducted in Northern Central Ethiopia [15], confidence interval of 95%, a margin of error of 5% and an expected non-response rate of 10%. Accordingly, the calculated sample size was 410. However, since the total number of all forms of TB cases in the study area is less than 10,000, we have considered a

finite population correction for sample size. This made the final sample size of study 234. Considering a non-response of 10% a total of 258 cases of TB was required.

To obtain a representative sample for this study, we selected six districts and two towns administrative randomly out of 13 districts and four towns administrative of the study areas. The total sample size was proportionately allocated for the 15 randomly selected health facilities based on the expected cases of TB patients who seek care at each health facility was undertaken after reviewing previous years' TB reports.

## Data collection tool and data collection procedure

A structured questionnaire suited from tools used in Tuberculosis prevalence surveillance, Addis Ababa, Ethiopia, and An in-depth analysis of TB patients pathway in the Eastern Mediterranean Region [19, 20] was used to gather the data. In addition, to draw clinical profiles of the patients from the TB registry, a data abstraction checklist was prepared. The questionnaire was initially prepared in English and then translated to the Amharic language, and translated back to English to check for any inconsistencies.

The questionnaire consisted of socio-demographic characteristics, clinical and health-seeking behavior, and knowledge about TB and related stigma. TB registration books were reviewed for tuberculosis diagnostic information, such as date of diagnosis, type of PTB, type of diagnostic investigation used to diagnose TB, nutritional status, HIV serostatus, patient category, and date of treatment initiation.

## Data quality assurance

To assure the data quality, the data collection tool was prepared after a review of relevant works of literature and similar studies. The training was given for one day both for data collectors and supervisors on briefing the general objective of the study, and discussing the contents of the questionnaire. Pre-testing of the tool was carried out on the 13(5%) of sample size outside my study area (Sodo Zuriya, Humbo Tabala HC) before starting the actual data collection and necessary corrections were made.

## Data management and analysis

The collected data were entered into Epi-data version 4.4.1 and exported to SPSS software version 20.0 for analysis. Data were summarized using frequency, proportions, mean, median, standard deviation, and inter-quartile range.

Patient delay days were further explored for skewness', kurtosis, normality plots (Q-Q plots and/or histograms), or Kolmogorov-Smirnov test to check for normality. Hence, the distribution of the number of days elapsed across different time points was not normal and median days were used as a cutoff point to define delays. Thus, the patient delay was defined based on median days elapsed between onset of illness to the first visit to the health facility. As the data were skewed, non-parametric tests (Mann–Whitney/Kruskal-Wallis) were employed to compare group differences in patients' delays. Mann–Whitney test was used to compare two groups and the Kruskal-Wallis test was used for comparing three or more groups.

Associations between the dependent variable (patient delay) and the independent variables were analyzed by calculating the Odds Ratios and 95% confidence interval. Independent variables with marginal associations ($P \leq 0.25$) in the bivariate analysis were entered in multivariate logistic regression analysis. The significant association of independent variables with the dependent variable was assessed by using a 95% confidence interval and a respective adjusted odds ratio (AOR). The logistic model's fitness was checked using Hosmer-Lemeshow GOF-test at p-value >0.05. A two-tailed-sided p-value of $\leq 0.05$ was taken as statistically significant.

## Operational definition and definition of terms

Patient delay periods were defined similarly to previous studies, and the median value was used as a cut-off value to make a simple comparison with previous similar studies.

**Patient delay.** The time interval (in days) between the initial onsets of the first symptoms of TB until the first visit to a formal health care provider. TB patients who consulted a formal health care provider longer than the median value after the onset of the initial constitutional signs and symptoms of TB were considered delayed [14, 15].

**The onset of tuberculosis symptom.** The time at which the first symptom (i.e. Cough and other constitutional symptoms like fever, weakness, and weight loss or chest pain) of the illness for which a patient's health care seeking began [21].

**Formal-health care providers:** modern government or private health care facilities such as clinics, health centers, and hospitals [15].

**Non-formal health providers:** These include traditional health providers, local injectors, and drug retail outlets [15].

**Knowledge about TB:** was assessed using eight items with "yes" or "no" questions including the cause of TB (microbe, bacteria, germ), TB is hereditary, TB is contagious, mode of TB transmission (breathing, sneezing, coughing, raw milk intake), symptoms of TB, TB is curable, length of treatment (6 month = yes, otherwise no) and TB treatment modalities as free = yes or for charge = no. Patients who scored more than the set average (50%) were considered knowledgeable and those who scored less than average were considered not knowledgeable [19].

**Food insecurity experience scale.** FIES is comprised of eight questions ranging in the severity of FI (Food insecurity) they measure, from low FI (question 1) to Severe Food insecurity (question 8). Respondents answer yes/no to the 8 questions and the responses are aggregated to give raw scores ranging from 0 to 8. FI was classified into 3 categories: 1) food secure (FS) with raw scores = 0–3; 2) moderate FI (MFI), with raw scores = 4–6; and 3) Sever FI, with raw scores = 7–8 [22].

## Ethical consideration

Ethical clearance was obtained from the Ethical review committee of Arba Minch University, College of Medicine and Health Science. Following the approval, an official letter of co-operation was written to concerned bodies by the Department of Public Health of Arba Minch University. Permission was also obtained from the Gamo Zone health department, district health office, and the respective health facilities. Informed verbal consent was obtained from each participant, after the necessary explanation about the purpose, procedures of the study, the importance of their participation, and their right to the decision of participating in the study. Participants were informed of their right to refuse to answer some or any of the questions, as well as the importance of maintaining the confidentiality of the information collected throughout the study by remaining anonymous, keeping their privacy by interviewing them in a separate room during the interview and locking records.

## Results

### Socio-demographic characteristics of the study participants

From two hundred fifty-eight TB patients invited, data of 255 TB patients were analyzed, excluding three patients for incompleteness of data giving the response rate of 99%. Accordingly, 161(63.1%) of the respondents were registered at health centers and 106(41.6%) were females. The median (IQR) age of the study participant was 25(21–28) years and the majority of the 112(43%) were among the age range of 18–24 years. One hundred forty (54.9%) of the

**Table 1. Socio-demographic characteristics of TB patients in Gamo zone public health facilities, Southern Ethiopia, 2019 (n = 255).**

| Variables | | Frequency | Percent (%) |
|---|---|---|---|
| Treatment center | Hospital | 94 | 36.9 |
| | Health center | 161 | 63.1 |
| Sex | Male | 149 | 58.4 |
| | Female | 106 | 41.6 |
| Age | 18–24 | 112 | 43.9 |
| | 25–44 | 97 | 38.0 |
| | ≥45 | 46 | 18.1 |
| Residence | Urban | 106 | 41.6 |
| | Rural | 149 | 58.4 |
| Educational status | Illiterate | 68 | 26.7 |
| | Primary school | 89 | 34.9 |
| | Secondary and above | 98 | 38.4 |
| Marital status | Single | 98 | 38.4 |
| | Married | 145 | 56.9 |
| | Widowed/Divorced | 12 | 4.7 |
| Family size | 1 to 3 | 63 | 24.7 |
| | >3 | 192 | 75.3 |
| Income status(US Dollar) | ≤ $13.91 | 159 | 62.4 |
| | $13.92–25.29 | 47 | 18.4 |
| | >$25.29 | 49 | 19.2 |
| Occupational status | Employed | 58 | 22.7 |
| | Farmer | 60 | 23.5 |
| | Student | 69 | 27.1 |
| | Unskilled worker[a] | 16 | 6.3 |
| | Unemployed/Housewife | 52 | 20.4 |
| Religion | Orthodox | 106 | 41.6 |
| | Protestant | 140 | 54.9 |
| | Others[b] | 9 | 3.5 |
| One way walking time | ≤30 min | 144 | 56.5 |
| | 30–60 min | 59 | 23.1 |
| | ≥60 min | 52 | 20.4 |

[a] Housemaid, daily laborer

[b] Muslim(6), Catholic(2), traditional(1)

cases were followers of protestant Christian and 149(58.4%) of the enrolled participant resided in rural. Concerning occupation and income, around 69 (27.1%) participants were students and 159(62.4%) belong to the income level of ≤ $13.91 respectively. The median time taken by patients from their home to initially visit the health facility was 30 minutes in one direction. (IQR 20–60). (**Table 1**)

## Health-care seeking behavior of tuberculosis patients

After the symptoms, 115 took action including self-treatment and use traditional medicine before the HCF visit. About the severity of the diseases at presentation, 144(56.5%) of the patients were ambulatory in functional status before contacting a formal health facility. Of all the respondents, 193(75.7%) perceived their first visit was delayed for which 170(66.7%) and

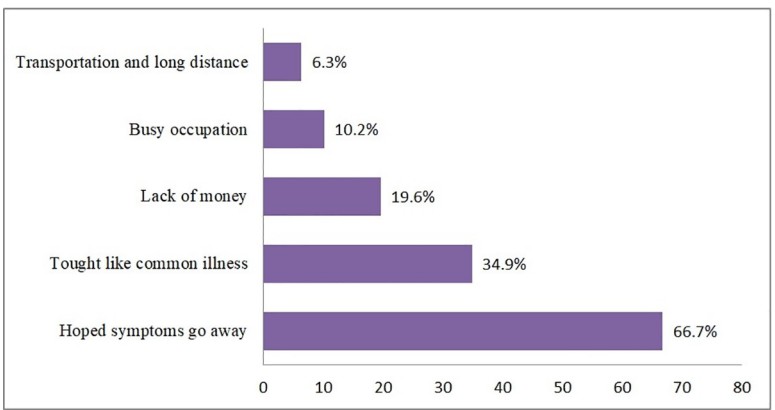

**Fig 1. Perceived reasons for delay health care seeking among TB patients in Gamo zone public health facilities, Southern Ethiopia, 2019.**

50 (19.6%) reasoned expecting the illness to limit by itself and lack of money respectively (**Fig 1**).

More than half 171 (67%) of patients first contacted public health facility and 214 (83.9%) were encountered more than one health care contacts before a diagnosis of TB made, and 17 (6.7%) patients were diagnosed at private facilities. Related to community-based health insurance (CBHI) status 44 (17.3%) were a member before seeking care among respondents (**Table 2**).

## Knowledge, behavioural and clinical characteristics of tuberculosis patient

Regarding knowledge about TB, 124 (48.6%) had relatively good knowledge about TB illness and its treatment. The majority of the TB patients knew that TB is curable 232(91%) and the duration of anti-TB treatment 155 (60.8%). Nearly 225(88.2%) of the patients knew that TB was a contagious disease. Regarding the level of stigma associated with TB, most 213(83.5%) of the respondents practiced High stigma on tuberculosis.

Of the total respondents who were assessed during diagnosis to check their nutritional status, 132 (51.8%) had a normal BMI of 18.5 to 24.99 kg/m2, while the rest were underweight. The majority of the respondents were in the food secure categories with respect to food insecurity experience (**Table 2**).

## Clinical characteristics of the participants at the presentation

At the onset of illness, the majority of patients came with a combination of symptoms. The most frequently reported symptom was cough in 209(82%) patients; followed by night sweating in 12(49.8%) patients, fatigue/weakness in 108(42.4%) patients, weight loss in 73 (28.6), chest pain in 74 (29.0) and loss of appetite in 70 (27.5) patients respectively (**Fig 2**).

More than half of the respondents 173 (67.8%) were smear-positive in classification and before the commencement of treatment, all of the cases were offered HIV screening tests of whom 21(8.2%) tested positive. Regarding contact history, in the last year with TB patients only 65(25.5%) had contact with TB patients.

## Delay period and associated factors

**Patient delay.** The median (IQR) days elapsed between onset of illness to first health facility visit (patient delay) was 30 (15–60). Of all recruited study participants, 145 (56.9%) did seek

**Table 2. Healthcare seeking behavior among TB patients in Gamo zone public health facilities, Southern Ethiopia, 2019 (n = 255).**

| Variables | | Frequency | Percent (%) |
|---|---|---|---|
| First action to illness | Visit HCF[a] | 126 | 49.4 |
| | Self-treatment | 92 | 36.1 |
| | Use traditional medicine | 22 | 8.6 |
| | Consult HEW[b] | 15 | 5.9 |
| Severity of disease at the 1st contact | Working | 88 | 34.5 |
| | Ambulatory | 144 | 56.5 |
| | Bedridden | 23 | 9.0 |
| Facility first visited | Private facilities | 64 | 25.1 |
| | Public hospital | 59 | 23.1 |
| | Health center | 112 | 43.9 |
| | Health post | 20 | 7.9 |
| Health care contacts | Single | 41 | 16.1 |
| | Multiple | 214 | 83.9 |
| Place of TB diagnosis | Public | 238 | 93.3 |
| | Private | 17 | 6.7 |
| CBHI status[c] | Yes | 44 | 17.3 |
| | No | 211 | 82.7 |
| Knowledge towards TB | Poor | 131 | 51.4 |
| | Good | 124 | 48.6 |
| TB associated Stigma | Low stigma | 42 | 16.5 |
| | High stigma | 213 | 83.5 |
| TB category | SPPTB[d] | 173 | 67.8 |
| | SNPTB[e] | 35 | 13.7 |
| | EPTB[f] | 47 | 18.5 |
| HIV status | Positive | 21 | 8.2 |
| | Negative | 234 | 91.8 |
| Contact history in the last 1 year | Yes | 65 | 25.5 |
| | No | 190 | 22.5 |
| BMI | Normal | 132 | 51.8 |
| | Underweight | 123 | 48.2 |
| FIES status[g] | Food secure | 124 | 48.6 |
| | Moderate food insecurity | 80 | 31.4 |
| | Sever food insecurity | 51 | 20.0 |

[a] health care facility

[b] Health Extension worker(trained females those provide a household package of health care to household)

[c] Community-based health insurance

[d] smear-positive pulmonary tuberculosis

[e] smear-negative pulmonary tuberculosis

[f] extrapulmonary tuberculosis

[g] Food insecurity experience scale

medical advice after 30 days of the onset of their illness. The median patient delay is significantly different with residence (p<0.001), TB patients from rural (median 30 days) longer patient delay than those from urban (median 21 days), educational status (p<0.001): Illiterate (median 45 days) and primary school (median 30 days) had longer patient delays than those secondary and above (median 21 days). Furthermore, type of TB (P = 0.03), occupational status (p = 0.002), one way walking time (p = 0.002), among type of symptom patients presenting

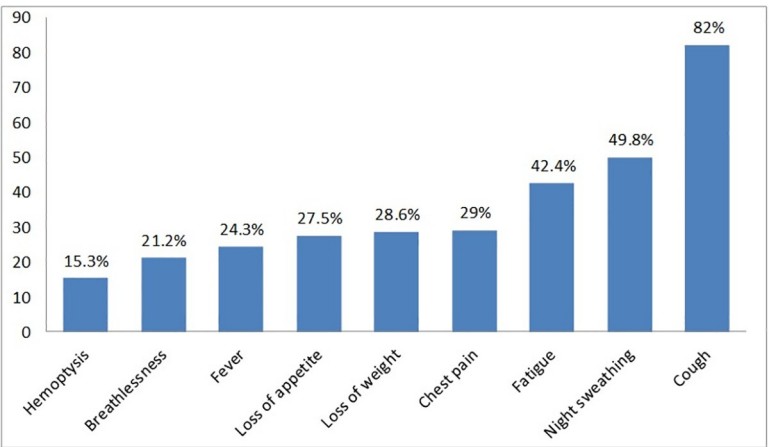

**Fig 2. Complaints, which made patients, seek medical care in Gamo zone public health facilities, Southern Ethiopia, 2019.**

with lose weight and appetite(p = 0.011 and p = 0.021 respectively), severity of disease at first contact (p = 0.006), first action (p = 0.001), facility first visited (0.024), number of health care contact (p = 0.043) knowledge towards TB (p = 0.004) were significantly associated with median patient delay (**Tables 3 & 4**).

On bivariate logistic regression, prolonged patients delay was significantly associated with rural residence, older age (≥45 years), having large family members, occupation, educational status, walking for more than 60 min to arrive at the health facility, not being a member of CBHI, smear-negative status, and EPTB, the severity of disease at the first contact, having symptoms of fever, loss of weight, chest pain, fatigue, loss of appetite and breathlessness, the first action of informal provider, the private facility first visited, having multiple health care contact, nutritional status of underweight, FIES of moderately and severe food insecurity and having high stigma. Nevertheless, having good knowledge of TB was significantly associated with reduced patients delay.

In multiple logistic regression, among types of symptoms: patient presetting with loss of weight (AOR: 2.53; 95%CI: 1.35,4.74) and fatigue (AOR: 2.38; 95%CI: 1.36,4.17), first action informal provider (AOR: 2.24; 95%CI: 1.29,3.86) and body mass index(BMI) categories of underweight (AOR: 1.74; 95%CI: 1.01,3.00) were independently associated with higher odds of patient delay beyond the median of 30 days. On the other hand having good knowledge about TB (AOR: 0.44; 95% CI; 0.26, 0.76) was independently associated with lower odds of patient delay beyond the median of 30 days (**Table 5**).

## Discussion

This study set out with the aim of assessing the magnitude and factors associated with patient delay among tuberculosis patients in Gamo Zone, Southern Ethiopia. The result of this study demonstrated that the median (IQR) of the patient delay was 30(15–60) days and the observed patient delay was agreed with previous studies in Ethiopia [15, 16, 23] and other low-and-middle-income countries (28–30 days) [24, 25]. The current study finding is lower than studies carried out in different areas of Ethiopia (36–63 days) [13, 26] and Nepal (50 days), and Ghana (59 days) [17, 27]. However, tuberculosis patients in China (median, 10 days) and Iran (median, 13 days) were substantially less delayed than those in this study [14, 28]. The cultural issue, low socioeconomic standing, and low level of knowledge and understanding of the

**Table 3. Distribution of patient delay by socio-demographic, clinical variables and health-seeking trajectories, non-parametric (Mann–Whitney and Kruskal-Wallis) test.**

| Characteristics | | Patient delay | |
|---|---|---|---|
| | | Median(IQR) | *p*-value |
| Total | | 30(15,60) | |
| Delayed, n (%) | | 145(56.9) | |
| Sex | Male | 30(15,60) | 0.672 |
| | Female | 30(15,60) | |
| Residence | Urban | 21(15,30) | **<0.001** |
| | Rural | 30(21,60) | |
| Age | 18–24 | 30(15,41.25) | 0.237 |
| | 25–44 | 30(15,60) | |
| | ≥45 | 30(15,60) | |
| Education status | Illiterate | 45 (21,90) | **<0.001** |
| | Primary school | 30(17.5,60) | |
| | Secondary and above | 21(15,30) | |
| Marital status | Single | 30 (15,30) | 0.244 |
| | Married | 30(15,60) | |
| | Widowed/Divorced | 30(15,52.5) | |
| Family size | 1 to 3 | 30(15, 30) | 0.166 |
| | >3 | 30(15,60) | |
| Occupational status | Employed | 30(15,30) | **0.002** |
| | Farmer | 45(21,90) | |
| | Student | 21(15,30) | |
| | Unskilled worker | 30(20,52.50) | |
| | Unemployed | 25.5(15,60) | |
| Income status | ≤ $13.91 | 30(15,60) | 0.094 |
| | $13.92–25.29 | 45(21,60) | |
| | >$25.29 | 30(15.37.5) | |
| CBHI[b] status | Members | 30(20,60) | 0.160 |
| | Not member | 30(15,60) | |
| One way walking time | <30min | 30(15,30) | **0.002** |
| | 30-60min | 30(15,45) | |
| | >60min | 30(21,90) | |

[a] Housemaid, daily laborer

[b] Community-based health insurance

disease are all potential explanations for disparities in delay. Besides that, the patients' prolonged delay in our study may be linked to their inability to recognize symptoms at the first onset of the disease. This result was supplemented by a recent national tuberculosis prevalence survey, which showed that less than half of TB patients meet the "TB suspect" criteria and of them have solely imprecise symptoms [29].

The proportion of patients who delayed beyond the median value of patients delay (30 days) was 56.9% in the current study, with a 95% CI (50.7%, 63.0%). Previous studies in Ethiopia [13, 30], other African countries [17, 25], and Pakistan [31] also reported consistent proportions.

However, the findings of this study were relatively higher than those of studies conducted in Ethiopia (31.3–46.3%) [13, 30], and other African countries Zimbabwe and Uganda [24,

**Table 4. Distribution of patient delay by socio-demographic, clinical variables and health-seeking trajectories, nonparametric (Mann–Whitney and Kruskal-Wallis) test** *(Continued).*

| Characteristics | | Patient delay | |
|---|---|---|---|
| | | Median(IQR[a]) | *p*-value |
| TB category | SPPTB[b] | 30 (15,60) | 0.552 |
| | SNPTB[c] | 21(15,45) | |
| | EPTB[d] | 30(21,60) | |
| Types of symptom | Cough | 30(15,60) | 0.698 |
| | Fever | 30(21,60) | 0.288 |
| | Loss weight | 30(21,60) | **0.011** |
| | Hemoptysis | 30(15,50) | 0.748 |
| | Chest pain | 30(20,60) | 0.136 |
| | Breathlessness | 30(21,60) | 0.072 |
| | Night sweating | 30(15,45) | 0.468 |
| | Fatigue | 30(21,60) | 0.072 |
| | Loss of appetite | 30(21,60) | **0.021** |
| Severity of disease at the 1st contact | Working | 21 (15,45) | **0.006** |
| | Ambulatory | 30(20,60) | |
| | Bedridden | 60(21,90) | |
| Contact history in the last 1 year | Yes | 30(21,60) | 0.662 |
| | No | 30(15,60) | |
| HIV[e] status | Positive | 21 (15,60) | 0.672 |
| | Negative | 30(15,60) | |
| First action | Informal provider | 30 (21,60) | **0.001** |
| | Formal provider | 21(15,30) | |
| Facility first visited | Government | 30 (15,45) | **0.024** |
| | Private | 30(21,60) | |
| Health care contacts | Single | 21(15,30) | **0.043** |
| | Multiple | 30(18.75,60) | |
| BMI[f] | Normal | 30(15,41.25) | 0.090 |
| | Underweight | 30(15,60) | |
| FIES[g] | Food secure | 30 (15,30) | 0.083 |
| | Moderate food insecurity | 30(15,60) | |
| | Sever food insecurity | 30(21,60) | |
| Knowledge towards TB | Poor | 30(21,60) | **0.004** |
| | Good | 21(15,45) | |
| TB associated stigma | Low stigma | 21(15,54) | 0.084 |
| | High stigma | 30(15,60) | |

*Note*: Using non-parametric Kruskal-Wallis test to compare three or more groups and Mann–Whitney test to compare two groups. Statistically significant values are in bold.

[a] interquartile range

[b] smear-positive pulmonary tuberculosis

[c] smear-negative pulmonary tuberculosis

[d] extrapulmonary tuberculosis

[e] human immunodeficiency virus

[f] Body Mass Index

[g] food insecurity experience scale.

**Table 5. Factors associated with patient delay among TB patients in Gamo zone public health facilities, Southern Ethiopia, bivariate and multivariate analysis, 2019.**

| Variables | | Patient delay(days) | | Crude and adjusted OR | |
|---|---|---|---|---|---|
| | | ≥30 | <30 | COR(95%CI) | AOR(95%CI) |
| Residence | Urban | 50(47.2) | 56(52.8) | 1 | 1 |
| | Rural | 95(63.8) | 54(36.2) | 1.97(1.19,3.27) | 1.64(0.94,2.86) |
| Age | 18–24 | 58(51.8) | 54(48.2) | 1 | 1 |
| | 25–44 | 58(59.8) | 39(40.2) | 1.38(0.80,2.40) | 1.25(0.56,2.81) |
| | ≥45 | 29(63.0) | 17(37.0) | 1.59(0.78,3.21) | 0.75(0.26,2.21) |
| Educational status | Illiterate | 46(67.6) | 22(32.4) | 2.27(1.19,4.32) | 1.55(0.56,4.27) |
| | Primary school | 52(58.4) | 37(41.6) | 1.52(0.85,2.72) | 0.90(0.43,1.95) |
| | Secondary and above | 47(48.0) | 51(52.0) | 1 | 1 |
| Occupation | Employed | 32(55.2) | 26(44.8) | 1 | 1 |
| | Farmer | 43(71.7) | 17(28.3) | 2.05(0.96,4.41) | 1.69(0.60,4.71) |
| | Student | 34(49.3) | 35(50.7) | 0.79(0.39,1.59) | 0.82(0.35,1.95) |
| | Unskilled worker | 10(62.5) | 6(37.5) | 1.35(0.43,0.42) | 0.80(0.21,3.12) |
| | Unemployed/Housewife | 26(50.0) | 26(50.0) | 0.81(0,38,1.72) | 0.74(0.29,1.93) |
| One way walking time | ≤30 min | 77(53.5) | 67(46.5) | 1 | 1 |
| | 30–60 min | 32(54.2) | 27(45.8) | 1.03(0.56,1.89) | 0.62(0.29,1.34) |
| | ≥60 min | 36(69.2) | 16(30.8) | 1.96(0.99,3.84) | 0.97(0.40,2.37) |
| CBHI[a] Status | Member | 30(68.2) | 14(31.8) | 1 | 1 |
| | Not member | 115(54.5) | 96(45.5) | 0.56(0.28,1.11) | 0.62(0.27,1.38) |
| TB category | SPPTB[b] | 102(59.0) | 71(41.0) | 1 | |
| | SNPTB[c] | 15(42.9) | 20(57.1) | 0.52(0.25,1.90) | 0.49(0.22,1.12) |
| | EPTB[d] | 28(59.6) | 19(40.4) | 1.03(0.53,1.98) | 1.41(0.65,3.06) |
| Severity of disease at the 1st contact | Working | 42(47.7) | 46(52.3) | 1 | 1 |
| | Ambulatory | 86(59.7) | 58(40.3) | 1.62(0.95,2.77) | 1.18(0.59,2.37) |
| | Bedridden | 17(73.9) | 6(26.1) | 3.10(1.12,8.61) | 1.71(0.45,6.56) |
| Types of symptoms | | | | | |
| Fever | Yes | 40(64.5) | 22(35.5) | 1.52(0.84,2.75) | 1.46(0.74,2.88) |
| | No | 105(54.4) | 88(45.6) | 1 | 1 |
| Loss of weight | Yes | 53(72.6) | 20(27.4) | **2.59(1.44,4.68)** | **2.53(1.35,4.74)**[*] |
| | No | 92(50.5) | 90(49.5) | 1 | |
| Chest pain | Yes | 47(63.5) | 27(36.5) | 1.47(0.84,2.57) | 1.44(0.73,2.82) |
| | No | 98(54.1) | 83(45.9) | 1 | 1 |
| Breathlessness | Yes | 35(64.8) | 19(35.2) | 1.52(0.82,2.84) | 1.34(0.67,2.68) |
| | No | 110(54.7) | 91(45.3) | 1 | 1 |
| Fatigue | Yes | 73(67.6) | 35(32.4) | **2.17(1.30,3.64)** | **2.38(1.36,4.17)**[*] |
| | No | 72(49.0) | 75(51.0) | 1 | |
| Loss of appetite | Yes | 48(68.6) | 22(31.4) | 1.98(1.10,3.54) | 1.60(0.80,3.21) |
| | No | 97(52.4) | 88(47.6) | 1 | 1 |
| First action | Informal provider | 78(68.4) | 36(31.6) | **2.39(1.43,4.00)** | **2.24(1.29,3.86)**[*] |
| | Formal provider | 67(47.5) | 74(52.5) | 1 | 1 |
| Facility first visited | Government | 102(53.4) | 89(46.6) | 1 | 1 |
| | Private | 43(67.2) | 21(32.8) | 1.78(0.99,3.24) | 1.52(0.75,3.04) |
| Health care contacts | Single | 18(43.9) | 23(56.1) | 1 | 1 |
| | Multiple | 127(59.3) | 87(40.7) | 1.86(0.95,3.66) | 113(0.46,2.80) |
| BMI[e] | Normal | 68(51.5) | 64(48.5) | 1 | |
| | Underweight | 77(62.6) | 46(37.4) | **1.57(0.96,2.60)** | **1.74(1.01,3.00)**[*] |

*(Continued)*

**Table 5.** (Continued)

| Variables | | Patient delay(days) | | Crude and adjusted OR | |
|---|---|---|---|---|---|
| | | ≥30 | <30 | COR(95%CI) | AOR(95%CI) |
| FIES[f] status | Food secure | 64(51.6) | 60(48.4) | 1 | 1 |
| | Moderate insecurity | 48(60.0) | 32(40.0) | 1.41(0.80,2.48) | 0.68(0.31,1.50) |
| | Sever food insecurity | 33(64.7) | 18(35.3) | 1.72(0.87,3.37) | 1.00(0.41,2.45) |
| Knowledge toward TB | Poor | 86(65.6) | 46(34.4) | 1 | 1 |
| | Good | 59(47.6) | 65(52.4) | **0.47(0.29,0.79)** | **0.44(0.26,0.76)***|
| TB associated stigma | Low stigma | 19(45.2) | 23(54.8) | 1 | 1 |
| | High stigma | 126(59.2) | 87(40.8) | 1.75(0.90,3.41) | 1.35(0.61,2.95) |

[a] Community based health insurance

[b] smear-positive pulmonary tuberculosis

[c] smear-negative pulmonary tuberculosis

[d] extrapulmonary tuberculosis

[e] Body Mass Index, food insecurity experience scale.

*$P$-value < 0.05; COR: Crude odds ratio; AOR: Adjusted odds ratio; CI: Confidence Interval; 1: Reference category

32], and lower than studies conducted in Bale Zone of Ethiopia's, where 89.9% of TB patients visited a health care provider after the median time had elapsed (30 days) [33]. Variations among these studies may be due to differences in countries' health systems, strategies and policies, and infrastructure, as well as differences in the sample population's socio-demographic features, such as rural versus urban settings, and pure agricultural versus pastoralist populations [34].

It is necessary to have adequate knowledge of tuberculosis in order to seek medical help as soon as possible. In this study, the extent of knowledge regarding TB was found to be significantly associated with patient delay. The study discovered that patients who have a good knowledge of TB disease and treatment are less likely to put off seeking care. Similarly, other studies have found that a lack of knowledge about tuberculosis and its treatment program as an explanation for the delay in seeking care [13, 26, 34, 35], and can endorse that lack of knowledge may additionally result in patients' reluctance in search of appropriate health care.

Using an informal care provider was found to be a strong predictor of patient delay during this study. Patients who went to non-formal health care providers first had longer patient delays than patients who went to formal health care providers first. Previous studies in Ethiopia [13, 26] and other African countries [35, 36] have reported consistent findings. This finding could be attributed to the fact that patients have taken several actions before visits to a formal health care facility that affect the timing of care-seeking. This could be ascribed to the use of such home remedies or over-the-counter antibiotics or analgesics, which could, for the time being, minimize the manifestation of the disease. Another potential reason for this result is that the relative abundance of drug stores provides an advantage of proximity as compared to diagnostic health facilities, as well as cost savings as drug stores do not charge for cards or laboratory services [13].

Following previous studies [37, 38], the most frequent symptoms in our patients were cough, night time sweating, fatigue, and chest pain. From the mentioned symptoms, weight reduction and fatigue have been found as a factor associated with prolonged patient delay. Accordingly, weight reduction was shown to be positively associated with a longer patient delay in multiple studies in Brazil, Melbourne, and Montenegro [27, 39, 40]. These findings may be clarified by the fact that patients perceive these symptoms as temporary signs of a

general disease, leading to self-treatment that lasts until deterioration and the appearance of more specific symptoms. Furthermore, timely referral to healthcare services for debilitating symptoms could also be difficult owing to financial constraints, low health awareness, and stigma [38]. However, this finding was contrary to the finding from Brazil [41]. A possible explanation for these differences may be the majority of the patient in Brazil might know that weight loss is one of TB symptoms.

Another finding from this study suggests that people with a BMI's in the underweight category are more presumably to delay than those with a normal BMI. These results further support the above finding that patient presenting with weight loss was more likely to delay than their counterpart. We also checked for Multicollinearity between weight loss and BMI's status, but there was no Multicollinearity [38]. However, this result has not previously been described. Those underweight patients possibly come from low socio-economic status, given that poverty commonly influences the health-seeking behavior of individuals.

## Conclusions

Our results demonstrated that a significant proportion of patients had delays that exceeded the suitable range. The median patient delay, according to our findings, was 30 days. About 56.9% of patients had prolonged patients' delay. The first action provided by an informal provider, a lack of knowledge about TB, underweight BMI status, and a patient presenting with symptoms of weight loss and fatigue were studied to be factors associated with prolonged patient delay.

The findings from this study suggest that the community should be sensitized to seeking appropriate health care as early as possible. The sensitization programs should take into consideration of different groups in a society such as women, elders, illiterate, and economically poor by using culturally convenient media of communication to ensure that the whole community is reached. Greater efforts are needed to ensure intensify TB case finding involving the community, formal and informal providers by decentralizing diagnostic and therapeutic services to lower-level public and private healthcare facilities. Health education sessions should be designed and provided to enhance accurate awareness dissemination on symptoms, medication options, and the curability of TB in the community, as well as patients attending primary health care facilities. There is no doubt that the maintenance of a high suspected tuberculosis index in the older population is justifiable. Given that different factors may delay the diagnosis of TB in elderly individuals, efforts should be made to reduce these delays in order to stop or control the spread of TB. Further research should be undertaken to investigate the in-depth understanding of reasons for a patient using qualitative designs and a community-based study can be done to capture symptomatic individuals who are not attending health facilities.

## Supporting information

**S1 Fig. Schematic representation of sampling procedure.**
(TIF)

**S1 Text. Additional definition and measurements.**
(DOCX)

**S1 File. Questionnaire.**
(PDF)

**S1 Data. The raw data supporting the finding of this article.**
(CSV)

## Acknowledgments

We are grateful to Arba Minch University, College of Health Sciences and Medicine for providing us with ethical clearance for this study. We also extend our appreciation to study participants, data collectors, Supervisors, and Health facility leaders.

## Author Contributions

**Conceptualization:** Asrat Arja, Wanzahun Godana.

**Data curation:** Asrat Arja, Wanzahun Godana, Biruk Bogale.

**Formal analysis:** Asrat Arja, Wanzahun Godana.

**Investigation:** Asrat Arja, Wanzahun Godana, Hadiya Hassen.

**Methodology:** Asrat Arja, Wanzahun Godana, Hadiya Hassen, Biruk Bogale.

**Project administration:** Asrat Arja.

**Resources:** Asrat Arja.

**Software:** Asrat Arja, Biruk Bogale.

**Supervision:** Asrat Arja, Wanzahun Godana, Hadiya Hassen.

**Validation:** Asrat Arja, Wanzahun Godana, Hadiya Hassen, Biruk Bogale.

**Visualization:** Asrat Arja.

**Writing – original draft:** Asrat Arja, Biruk Bogale.

**Writing – review & editing:** Asrat Arja, Wanzahun Godana, Hadiya Hassen, Biruk Bogale.

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
