## [Decision Letter · Decision Letter 0]

19 Jan 2021

PONE-D-20-27080

PATIENT DELAY AND ASSOCIATED FACTORS AMONG TUBERCULOSIS PATIENTS IN GAMO ZONE PUBLIC HEALTH FACILITIES, SOUTHERN ETHIOPIA; 2019:AN  INSTITUTION-BASED CROSS-SECTIONAL STUDY

PLOS ONE

Dear Dr. Arja,

Thank you for submitting your manuscript to PLOS ONE. After careful consideration, we feel that it has merit but does not fully meet PLOS ONE’s publication criteria as it currently stands. Therefore, we invite you to submit a revised version of the manuscript that addresses the points raised during the review process.

We look forward to receiving your revised manuscript.

Kind regards,

Claudia Marotta

Academic Editor

PLOS ONE

Additional Editor Comments:

dear authors follow reviewer suggestion to improve your article

2. Please include additional information regarding the survey or questionnaire used in the study and ensure that you have provided sufficient details that others could replicate the analyses. For instance, if you developed a questionnaire as part of this study and it is not under a copyright more restrictive.

3. Please upload a copy of Figure 4, to which you refer in your text on page 8. If the figure is no longer to be included as part of the submission please remove all reference to it within the text.

4. Please ensure that you refer to Figure 1 in your text as, if accepted, production will need this reference to link the reader to the figure.

5. We note you have included tables to which you do not refer in the text of your manuscript. Please ensure that you refer to Tables 1, 3, and 5 in your text; if accepted, production will need this reference to link the reader to the Table.

7. Thank you for submitting the above manuscript to PLOS ONE. During our internal evaluation of the manuscript, we found significant text overlap between your submission and the following previously published works:

https://bmcpublichealth.biomedcentral.com/articles/10.1186/s12889-018-5823-9?

https://bmcinfectdis.biomedcentral.com/articles/10.1186/s12879-019-4089-x

https://bmcinfectdis.biomedcentral.com/articles/10.1186/s12879-019-4089-x

Please revise the manuscript to rephrase the duplicated text, cite your sources, and provide details as to how the current manuscript advances on previous work. Please note that further consideration is dependent on the submission of a manuscript that addresses these concerns about the overlap in text with published work.

Reviewers' comments:

Reviewer's Responses to Questions

**Comments to the Author**

1. Is the manuscript technically sound, and do the data support the conclusions?

Reviewer #1: Yes

Reviewer #2: Yes

Reviewer #3: Yes

2. Has the statistical analysis been performed appropriately and rigorously? 

Reviewer #1: Yes

Reviewer #2: Yes

Reviewer #3: Yes

3. Have the authors made all data underlying the findings in their manuscript fully available?

Reviewer #1: Yes

Reviewer #2: Yes

Reviewer #3: Yes

4. Is the manuscript presented in an intelligible fashion and written in standard English?

Reviewer #1: Yes

Reviewer #2: No

Reviewer #3: Yes

5. Review Comments to the Author

Reviewer #1: Acceptable findings for a common serious disease in an endemic country. Both Introduction and Methods are narrated in a scientific way. Results flowed smoothly and conclusions met the study objectives.

Reviewer #2: The manuscript is very interesting but it is too long: all the sections must be summarized. The main section are the RESULTS. There are tables with frequencies and other with the results related with these frecuencies and the text related.

Reviewer #3: In general, although not unprecedented, the article makes relevant contributions to the understanding of access to TB diagnosis and treatment in an important region of Ethiopia.

In the section “Introduction”: in the fourth paragraph that reads “Thus, Ethiopia has been listed among the 14 TB, TB / HIV (Human Immunodeficiency Virus) and Multi-Drug Resistant TB (MDR TB) high burden Countries (HBC) that accounted for 80% of all estimated TB cases Worldwide (7) ”. The reference (7) cites 30 high burden countries: 20 by absolute number of TB cases plus 10 based on severity of disease burden - incidence per capita. In the last paragraph of the same section where it reads “Therefore, the aim of this study is to assess the magnitude and factors associated with patient and health system delays among tuberculosis patients in Gamo Zone public health facilities”, the results and discussion do not explore the factors associated with the delay associated to the health system. Therefore, I suggest removing the delay related to the health system from the aim of the study.

Material and methods: it is not clear how the patients were selected and recruited, if there was sampling for smear positive and smear negative patients. Since smear negative at diagnosis may be related to the early search for care, I suggest having a separate analysis for smear negative patients; the same applies to extrapulmonary TB, in which the search for diagnosis can take longer because the symptoms are often more insidious. In cases of smear negative patients, how was TB diagnosis confirmed?

In the item “Study setting and design” the first paragraph describes the health network of Gamo Zone and there is no mention of the “health posts” that appear in the second paragraph. I suggest clarifying what "health posts" are and make up the health network.

In the presentation of the “Results”, Table 1 describes that 9 people had other religions and the note in the same Table referring to the letter “b” describes “b Muslim (6), Catholic (10), traditional (17)”. What are the numbers in parentheses? In this section I suggest dollarizing the value of "income status" to give an idea of the degree of poverty in relation to international parameters. For example, in 2015 the United Nations Organization classified the income of USD 1.90 / day / person as extreme poverty. It is also not clear whether the income was per family or per capita.

At the end of the fourth paragraph of the “Discussion” the authors state: “Another possible explanation for this finding is that the relative abundance of drug store offers an advantage of proximity compared to diagnostic health facilities and the drug store also have added advantages of cost minimization, as they do not charge for cards and laboratory services (17)"

however, in the“ Study setting and design” section we can read “Patients have free access to TB diagnosis and treatment in public health facilities ”. The sentences seem contradictory. I suggest that the authors clarify.

In the “Conclusion” section there is text in red. At the end of the text it reads “(review it)”. What does that mean? In addition, one of the conclusions is that there is a need for “regular refresher trainings”, but there are no data related to “health system delays” in the results or in the discussion section that can suspport this statement. I suggest removing from the last paragraph of the “introduction” section the “factors associated .... health system delays”, as already pointed out above.

6. PLOS authors have the option to publish the peer review history of their article (what does this mean?). If published, this will include your full peer review and any attached files.

Reviewer #1: **Yes: **Layth Al-Salihi

Reviewer #2: **Yes: **M.N. Altet Gomez

Reviewer #3: No

---

## [Author Response · Author response to Decision Letter 0]

17 Apr 2021

Dear Editors and Reviewers,

Thank you for giving us the opportunity to submit a revised draft of our manuscript titled “Patient delay and associated factors among tuberculosis patients in Gamo zone public health facilities, Southern Ethiopia: An institution-based cross-sectional study” to PLOSE ONE. We appreciate the time and effort that you and the reviewers have dedicated to providing your valuable feedback on our manuscript. We are grateful to the reviewers for their insightful comments on our paper. We have been able to incorporate changes to reflect most of the suggestions provided by the reviewers.

---

## [Decision Letter · Decision Letter 1]

15 Jul 2021

Patient delay and associated factors among tuberculosis patients in Gamo zone public health facilities, Southern Ethiopia: An institution-based cross-sectional study

PONE-D-20-27080R1

Dear Dr. Arja,

We’re pleased to inform you that your manuscript has been judged scientifically suitable for publication and will be formally accepted for publication once it meets all outstanding technical requirements.

Kind regards,

Marian Loveday, Ph.D.

Academic Editor

PLOS ONE

Reviewers' comments:

1. If the authors have adequately addressed your comments raised in a previous round of review and you feel that this manuscript is now acceptable for publication, you may indicate that here to bypass the “Comments to the Author” section, enter your conflict of interest statement in the “Confidential to Editor” section, and submit your "Accept" recommendation.

Reviewer #1: All comments have been addressed

Reviewer #2: All comments have been addressed

Reviewer #3: All comments have been addressed

2. Is the manuscript technically sound, and do the data support the conclusions?

Reviewer #1: Yes

Reviewer #2: Yes

Reviewer #3: Yes

3. Has the statistical analysis been performed appropriately and rigorously? 

Reviewer #1: Yes

Reviewer #2: Yes

Reviewer #3: N/A

4. Have the authors made all data underlying the findings in their manuscript fully available?

Reviewer #1: Yes

Reviewer #2: Yes

Reviewer #3: Yes

5. Is the manuscript presented in an intelligible fashion and written in standard English?

Reviewer #1: Yes

Reviewer #2: Yes

Reviewer #3: Yes

6. Review Comments to the Author

Reviewer #1: The authors used a scientific language in expressing their study. The article context is kept in line the aim of the study.

Reviewer #2: The revision has been well adapted to the reviewers comments. The Delay in diagnostic of Tuberculosis is an important issue in all TB programmes.

Reviewer #3: (No Response)

---

## [Editor Report · Acceptance letter]

22 Jul 2021

PONE-D-20-27080R1 

Patient delay and associated factors among tuberculosis patients in Gamo zone public health facilities, Southern Ethiopia: An institution-based cross-sectional study 

Dear Dr. Arja:

I'm pleased to inform you that your manuscript has been deemed suitable for publication in PLOS ONE. Congratulations! Your manuscript is now with our production department. 

Kind regards, 

on behalf of

Dr. Marian Loveday 

Academic Editor

PLOS ONE